# Trends in Research on Exosomes in Cancer Progression and Anticancer Therapy

**DOI:** 10.3390/cancers13020326

**Published:** 2021-01-17

**Authors:** Dona Sinha, Sraddhya Roy, Priyanka Saha, Nabanita Chatterjee, Anupam Bishayee

**Affiliations:** 1Department of Receptor Biology and Tumour Metastasis, Chittaranjan National Cancer Institute, Kolkata 700 026, India; sraddhya1@gmail.com (S.R.); poojasaha.saha79@gmail.com (P.S.); nabanita.chatterje@yahoo.com (N.C.); 2Lake Erie College of Osteopathic Medicine, Bradenton, FL 34211, USA

**Keywords:** tumor-derived exosomes, exosomal cargoes, protumorigenic effect, drug resistance, anticancer therapy

## Abstract

**Simple Summary:**

Intensive research in the field of cancer biology has discovered a unique mode of interplay between cells via extracellular bioactive vesicles called exosomes. Exosomes serve as intermediators among cells via their cargoes that, in turn, contribute in the progression of cancer. They are ubiquitously present in all body fluids as they are secreted from both normal and tumor cells. These minuscules exhibit multiple unique properties that facilitate their migration to distant locations and modulate the microenvironment for progression of cancer. This review summarizes the multifarious role of exosomes in various aspects of cancer research with its pros and cons. It discusses biogenesis of exosomes, their functional role in cancer metastasis, both protumorigenic and antitumorigenic, and also their applications in anticancer therapy.

**Abstract:**

Exosomes, the endosome-derived bilayered extracellular nanovesicles with their contribution in many aspects of cancer biology, have become one of the prime foci of research. Exosomes derived from various cells carry cargoes similar to their originator cells and their mode of generation is different compared to other extracellular vesicles. This review has tried to cover all aspects of exosome biogenesis, including cargo, Rab-dependent and Rab-independent secretion of endosomes and exosomal internalization. The bioactive molecules of the tumor-derived exosomes, by virtue of their ubiquitous presence and small size, can migrate to distal parts and propagate oncogenic signaling and epigenetic regulation, modulate tumor microenvironment and facilitate immune escape, tumor progression and drug resistance responsible for cancer progression. Strategies improvised against tumor-derived exosomes include suppression of exosome uptake, modulation of exosomal cargo and removal of exosomes. Apart from the protumorigenic role, exosomal cargoes have been selectively manipulated for diagnosis, immune therapy, vaccine development, RNA therapy, stem cell therapy, drug delivery and reversal of chemoresistance against cancer. However, several challenges, including in-depth knowledge of exosome biogenesis and protein sorting, perfect and pure isolation of exosomes, large-scale production, better loading efficiency, and targeted delivery of exosomes, have to be confronted before the successful implementation of exosomes becomes possible for the diagnosis and therapy of cancer.

## 1. Introduction

Exosomes are bilayered endosomal nanovesicles, first discovered in 1983, as transferrin conjugated vesicles (50 nm) released by reticulocytes [1]. Due to the increasing interest of scientists in exosome biology, a consensus guideline was proposed by board members of International Society of Extracellular Vesicles under “minimal experimental requirements for definition of extracellular vesicles and their functions” (MISEV2014) which was later updated in 2018 (MISEV2018). The guidelines advocated norms for nomenclature, isolation, separation, characterization, functional studies, and reporting requirements for proper identification of and experimentation with extracellular vesicles and exosomes [2,3]. Exosomes are generally formed by inward budding of late endosomes, also known as multivesicular bodies (MVBs). Intraluminal vesicles (ILVs) of MVBs engulf a variety of biomolecules which are released into extracellular space as exosomes. Exosomes are anucleated particles naturally released by cells, surrounded by lipid bilayer and are not capable of replication. Exosomes are identified by size (30–200 nm) and surface markers, such as membrane-associated proteins, e.g., lysosome-associated membrane glycoprotein 3 (LAMP3)/CD63; intercellular adhesion molecule (ICAM1)/CD81; and tetraspanin membrane protein/CD9. Exosomes are observed in various body fluids, such as blood, plasma, saliva, urine, synovial fluid, amniotic fluid, and breast milk [4,5].

All cellular types (normal and diseased) secrete exosomes, mediating intercellular communications [6]. Exosomes exhibit heterogeneity in size—Exo-Large (90–120 nm), Exo-Small (60–80 nm), and the membrane-less exomere (<50 nm). Exosome-mediated intercellular transfer of specific repertoire of proteins, lipids, RNA and DNA confer physiological and/or pathological functions to the recipient targets. Exosomes regulate physiological functions, such as neuronal communication, immune responses, reproductive activity, cell proliferation homeostasis, maturation and cellular waste disposition. They also contribute in clinical disorders, including inflammation, cancer, cardiovascular diseases, neuronal pathologies and pathogenic infections [5].

Our review deals with exosomal contents, exosome-associated protumorigenic, antitumorigenic effect and therapeutics, unlike other reviews, which discuss combinational roles of all microvesicles in cancer progression [7,8] or have primarily focused on tumor-derived exosomes (TEXs) with little information on therapeutics [9]. In contrast to reviews which have focused on specific exosomal cargoes and therapeutics [10,11], we have envisaged the exosomal contents, the mechanisms influencing cancer progression and their therapeutic implications in cancer management. The inexplicable nature of exosomes has raised concern about their role in the invasion and metastasis of cancer cells, encompassing epithelial-to-mesenchymal transition (EMT), angiogenesis, and immune regulation [12]. Thus, instead of reviewing the isolated impact of exosomes, e.g., evasion of immune surveillance [13] for cancer progression, we have tried to encompass exosome-mediated propagation of oncogenic signaling, epigenetic regulation, modulation of tumor microenvironment (TME) and immune escape, EMT, angiogenesis, metastasis and drug resistance. Considering the clinical applications, the exosomes serve as potent diagnostic and prognostic biomarkers because of their bioavailability, low toxicity and differentiated surface markers [5]. Recent reviews on exosomes have focused on therapeutic efficacy of exosomes by addressing extracellular vesicular interaction with the host immune system [14], constraints and opportunities available with bioengineering of exosomes [15,16,17], success against multiple cancers [18] and exosome-based drug delivery [19,20,21]. Anticancer treatments sometimes experience shortfall in their efficacy due to unwanted side effects of the therapeutic agents or shortened shelf-life, but exosomes serve as natural agents to overcome these issues and become a potent therapeutic agent [22]. However, instead of perceiving specific therapeutic potential of exosomes, the present review has tried to decipher the entire repertoire of exosomes, including both protumorigenic and antitumorigenic impact.

## 2. Cargo Composition of Exosomes

Exosomes are rich in enzymes, transcription factors, heat shock proteins (Hsps), major histocompatibility complex (MHC), cytoskeleton components, signal transducers, tetraspanins, lipids, RNAs and DNAs [6,23]. Detailed information about the exosomal components can be accessed via databases, such as ExoCarta [www.exocarta.org], EVpedia [http://evpedia.info] and Vesiclepedia [www.microvesicles.org]. Though exosomes diverge in size and biomolecular inclusions, some common components are observed in all types [5]. Lipid components are cholesterol, sphingomyelin, glycosphingolipids, phosphatidylcholine, phosphatidylserines, phosphatidylethanolamines and saturated fatty acids [4]. RNAs include specific microRNAs (miRNAs), long non-coding RNAs (lncRNAs), vault RNA, Y-RNA, transfer RNAs (tRNAs), ribosomal RNA (rRNA) fragments (such as 28S and 18S rRNA subunits) and messenger RNAs (mRNAs) [24]. Exosomal cargo components also include mitochondrial DNA (mtDNA), single-stranded DNA, double-stranded DNA and retrotransposons [4,6]. Different protein forms include components of the immune system (MHC class I and II molecules, cytokines), endosomal sorting complexes required for the transport (ESCRT) complex, those involved in trafficking (tetraspanins, glycosylphosphatidylinositol-anchored proteins, Rabs, soluble N-ethylmaleimide-sensitive fusion protein attachment protein receptors (SNARES), flotillins, lipid-rafts residents [25] and those involved in carcinogenesis (oncoproteins, tumor suppressor proteins, and transcriptional factors) [4]. The plasma membrane (PM) proteins constitute the vesicle membrane for maintaining composition parity with the cell membrane which helps in sequestration of soluble ligands. Exosomal proteins are involved in (i) antigen presentation, (ii) cell adhesion, (iii) cell structure and motility, (iv) stress regulation, (v) transcription and protein synthesis, and (vi) trafficking and membrane fusion [26]. The structure of exosome with membrane proteins and cargoes have been depicted in Figure 1.

## 3. Exosome Biogenesis

Endocytosis generates early endosomes via invagination of PM rich in lipid rafts. This internalizes the PM receptors which are either recycled or degraded. The exosome biogenesis involves a complex network of enzymatic actions and signal transductions. Early endosomes mature to MVBs or late endosomes upon internal budding of endosomes, forming ILVs [23]. MVB budding is primed with actin polymerization at PM lipid domains [27,28]. ADP ribosylation factor 6 (ARF6), along with phospholipase D2 (PLD2), converts ILVs into mature MVBs [29]. Heparanase enzyme stimulates the syndecan-syntenin-ALG-2 interacting protein X (ALIX) axis, upregulating exosome formation [30]. ARF6-induced actomyosin contractility and ESCRTs promote ILVs shedding from MVBs as exosomes [31]. The MVBs undergo one of the three type consequences [23,32] mentioned below:(i)Recycling through the trans-Golgi network (TGN) which may be subdivided into a fast and a slow pathway, considering the duration taken by the specific proteins/lipids from internalization to re-exposure at the cell surface or exocytosis.(ii)Lysosomal degradation by hydrolytic enzymes which are able to digest complex macromolecules.(iii)Fusion of MVBs with the cell surface release exosomes via exocytosis. Additional materials may be incorporated to the TGN at any juncture and processed through the canonical secretory pathways.

## 4. Sorting of Exosomal Cargoes

### 4.1. ESCRT-Dependent Sorting Pathway

The ESCRT pathway participates in sorting ubiquitinated proteins of exosome, after being internalized within ILVs. The complex includes ESCRT-0, which identifies and processes ubiquitin-dependent cargo inside the vesicles; ESCRT-I and ESCRT-II evoke budding and ESCRT-III causes vesicle scission from endosomal membrane. Other accessory proteins such as ALIX aid in vesicle budding and vacuolar protein sorting associated protein 4 (VPS4) promotes scission [30,33].

### 4.2. ESCRT-Independent Exosomal Sorting

Ceramide and cholesterol, PLD2, or tetraspanins mediates ESCRT independent sorting machinery. Tetraspanins may promote incorporation of specific cargoes into exosome, e.g., CD9 facilitates encapsulation of metalloproteinase CD10 and CD63. Even the lipid composition and membrane dynamics of the early endosome and MVBs may regulate exosomal cargoes. Ceramide and neutral sphingomyelinase 2 (nSMase2) play a pivotal role in an ESCRT independent process of exosome formation, loading, and release [23]. Podoplanin, a transmembrane glycoprotein, is another regulator of exosome biogenesis and cargo sorting [31].

## 5. Exocytosis and Secretion of Exosomes

Exocytosis is exosomal secretion into the extracellular matrix (ECM) which is regulated by Rab GTPases, molecular motors, cytoskeletal proteins, SNAREs, intracellular Ca^2+^ levels (increased Ca^2+^ results in increased exosome secretion) and extracellular/intracellular pH gradients [23]. Vesicular SNAREs (v-SNARE) on the MVB bind with the target SNARE (t-SNARE), Syx 5, on the inner surface of the PM for mediating fusion of MVB with the cell membrane [34]. The fusion of exosome with PM occurs at the actin-rich zones of the invadopodia, promoting ECM degradation and metastasis, followed by their exocytosis into extracellular space [34]. Peptidyl arginine deiminases aid exosomal secretion by deaminating actin [35]. A negative feedback mechanism limits excess exosome secretion from the same cells [34].

### Rabs Control Endocytic Pathway

The Rab GTPases belong to a large family of highly conserved proteins with 60 members, which regulate vesicular trafficking in eukaryotes. Different Rab forms are involved in endocytic trafficking—Rab4, 5, 9, 11a, 11b, 25 and 35 control recycling [36,37,38,39]; Rab5 and 7 cause endosomal maturation [40]; Rab 7 regulates sorting and degradation [41]; Rab 7, 27a and b control secretion of exosomes [42,43] and Rab5 overexpression causes release of exosomal markers [44]. Deregulation of the Rabs perturb the progression of cargo at specific endocytic locations. Rabs also play a crucial role in the regulation of tumor-derived exosomes. Rab11 influenced extrusion of exosome and interaction of MVB with autophagosomes [45] and promoted calcium dependent docking of MVBs to the PM [46] in K562 cells. Rab27A, in association with its GTPase activator, EP164, promoted exosome secretion by A549 lung cancer cells [47]. Rab27A/B are associated with exchange of exosomes between different cells of TME as well as with exosome secretion by macrophages [6]. Various types of Rabs involved in endocytic cargo trafficking have been depicted in Table 1.

## 6. Exosomal Internalization by Recipient Cells

Exosomes float in the ECM after their release and exosomal surface proteins help in detecting the target cells for their internalization [48]. Exosomes attach to specific target cells by receptor-ligand binding, mediated through integrins, tetraspanins and intercellular adhesion molecules, which then internalizes exosomes (Figure 2) by (i) clathrin/caveolin-mediated endocytosis, (ii) uptake via lipid raft, (iii) macropinocytosis, (iv) direct fusion with the PM and (v) phagocytosis.

Clathrin protein forms a mesh like structure around the exosomes for its internalization. The PM of the recipient cells forms an inward invagination, followed by pinching off the clathrin coated vesicle from the membrane. The exosome empties all its contents in recipient cell’s endosomes to perform specific functions [49]. Endocytosis, similar to the clathrin-dependent process, may be also mediated by caveolin-1 whose aggregations in PM form rafts. The invagination of the PM (caveolae) is rich in glycolipids, cholesterol and caveolin 1 [50]. Macropinocytosis involves distortion of PM forming protrusions from the membrane which encompass a region of extracellular fluid and exosomes, thereby internalizing exosomes. This process is Rac1-, actin- and cholesterol-dependent and it requires Na^+^/H^+^ exchange [51]. LAMP-1, integrins or tetraspanins are involved in the fusion of exosomes with the PM of recipient cells [52,53]. Phagocytosis is similar to macropinocytosis where exosomes are internalized along with some extracellular fluids. This process is followed by both phagocytic cells—like macrophages and dendritic cells (DCs)—and non-phagocytic cells like γδ T cells [54]. During exosome uptake by soluble signaling, exosomal ligands are cleaved by cytoplasmic proteases and are bound to their respective receptors present on the PM of the recipient cells. In case of juxtacrine signaling, the ligands and receptors need to be in close proximity for efficient ligand–receptor binding [55]. Exosomal tetraspanins (CD9, CD63, CD81 and CD82) regulate cell fission and fusion, target cell selection [42], migration, adhesion, proliferation, and interaction between exosomes and recipient cells [56]. Size distribution in exosomes facilitates their internalization since cells have a propensity for loading smaller exosomes [5]. Oncogenic integrins play a dominant role during internalization of tumor-derived exosomes by recipient cells. Metastasis has been observed to be associated with exosome-integrins, such as αvβ6 integrin in prostate, αvβ5 integrins in liver and α6β4 and α6β1 integrins in lung [56].

## 7. TEX

The TEXs influence shaping of the TME, tumor progression, invasion and premetastatic niche formation, metastasis, angiogenic switch, and immune escape by paracrine subversion of local and distant microenvironments [57].

### 7.1. Oncogenic Signaling Involved in Exosomal Trafficking

According to the genometastatic theory, complex biomolecules in exosomes transfer oncogenic traits to target cells. Matrix cells in the TME interact with their oncogenic counterparts through exosomes and mediate tumor evolution and progression. Exosomal cargoes confer oncogenic transformation, EMT, immune surveillance evasion, invasion, and metastatic properties to the recipient cells [58]. Hypoxia and extracellular acidity culminate in greater release of TEXs [58]. Cells having even one oncosuppressor mutation are more prone towards uptake of exosomal oncogenic factors. Mutations leading to upregulated mitogen-activated protein kinase (MAPK) signaling in cancer cells elevated exosomes release [59]. Secretion of exosomes by activated platelets promoted MAPK and phosphoinositide 3-kinase (PI3K)/protein kinase B (Akt)/matrix metalloproteinase (MMP) signaling during cancer progression [31]. Expression of oncogenic RAS in non-tumorigenic epithelial cells promoted secretion of oncoprotein-rich exosomes [60]. Robust expression of oncogenic and truncated forms of epidermal growth factor receptor (EGFR) vIII in glioma cells augmented exosomal secretion and transfer of oncogenic activity to other normal cells [61]. Mutation of liver kinase B1 (STK11), a tumor suppressor, increased exosome secretion in lung cancer [62]. Secretion of exosomal mtDNA induced anaerobic metabolism and dormancy in cancer cells [31].

### 7.2. Exosomal miRNA-Mediated Cancer Promotion

Breast TEXs, enriched with Dicer, Protein Argonaut 2, and transactivation response element RNA-binding protein, processed precursor miRNAs into mature miRNAs for gene silencing in target cells and induced non-tumorigenic epithelial cells to form tumors [63]. Exosomal miRNAs suppressed cell proliferation by downregulating the C-X-C motif chemokine ligand 12 (CXCL12); exosomal-miR-23b augmented cell quiescence by inhibiting myristoylated alanine-rich C-kinase substrate expression in the metastatic niche [64]; miR-10b molded the TME to promote tumor metastasis [65] of breast cancer (BC) cells. Astrocyte-derived exosomes suppressed phosphatase and tensin homolog (PTEN) by intracellular trafficking of miR-19a in metastatic BC and melanoma brain metastasis models [66]. Release of exosomal miR-1245 from mutant p53 cancer cells reoriented macrophages to transforming growth factor-β (TGF-β)-rich tumor-associated macrophages (TAMs) which, in turn, propagated tumor progression [67]. Exosomal miR-105 and miR-939 in BC and miR-181c in brain cancer dissolved tight junctions, caused vascular leakiness and induced metastasis [31].

### 7.3. Exosomes and TME

TEXs are well documented for immune suppression by multiple interactions with immune cells of the TME (Figure 3). They hinder helper and cytotoxic T-cell activation and function, activate regulatory T-cell (Tregs), inhibit cytotoxicity of natural killer (NK) cells, augment differentiation of myeloid-derived suppressor cells (MDSCs) and reduce leukocyte adhesion [34]. Exosomes modulate the TME by extracellular signal-regulated kinase (ERK)-mediated cell growth or apoptosis. Interaction of stromal cells and tumor via exosomes inflict dissemination of tight junctions, generating a suitable niche for metastasis [68]. TEXs induced cancer-associated fibroblasts (CAFs) for exosomes’ release [69]. The transfer of CAF-derived exosomal cargoes in the form of metabolic intermediates of the tricarboxylic acid cycle to cancer cells promotes neoplastic growth by alteration of glycolysis and glutamine-dependent reductive carboxylation [70]. Exosomes transformed fibroblasts into CAFs in melanoma [71]. CAFs or mesenchymal stem cells (MSCs) derived exosomes maneuvered Wnt signaling-induced migration [68]. Exosomes expressing Fas ligand activated CD8+ T-cell apoptosis [72]. Exosomal αvβ6 integrin inhibited the signal transducer and activator of transcription 1 (STAT1)/MX1/2 signaling in cancer cells and reprogramed monocytes into the M2 phenotype [73]. Exosomal miR-146a-5p from hepatocellular carcinoma (HCC) cells induced M2 polarization [74]. BC cell derived exosomes inhibited NK cells [75] and infiltrated neutrophils into tumors [76]. Melanoma-derived exosomes perturbed maturation of DCs in lymph nodes [77]. However, TEXs can supply antigens to DCs for cross-presentation to cytotoxic T cells [78]. Administration of topotecan/radiation induced the release of exosomal immunostimulatory DNA, which inflicted DC maturation and cytotoxic T cell activation [31]. Programmed death ligand 1 (PD-L1)-positive exosomes positively correlated with head and neck squamous cancer cells (HNSCC) progression in patients and administration of anti-PD-L-1 antibodies inhibited the immunosuppressive function of PD-L1 [79].

### 7.4. Impact of Exosomes on EMT, Invasion, Metastasis and Angiogenesis

Exosomal cargoes CD151 and Tspan8 are related with ECM degradation, stromal reprogramming, cell motility and tumor progression [80]. EMT was induced by exosomal miR-663b in bladder cancer [81]; lncRNA SOX2 overlapping transcript (Sox2ot) in pancreatic ductal adenocarcinoma (PDAC cells) [82]; and TGF-β-enriched TEXs in myofibroblasts [83]. Migration of tumor cells was facilitated by the exosome-mediated transfer of αvβ6 in prostate cancer [84]; miR-21 in bladder cancer [85]; TAM derived exosomes in gastric cancer (GC) cells [86]; and lncRNA ubiquitin-fold modifier conjugating enzyme 1 (UFC1) in non-small cell lung carcinoma (NSCLC) [87]. Exosomal lncRNA zinc finger antisense 1 (ZFAS1) induced EMT and migration in GC cells [88]. Metastasis was promoted by exosomal EGFR in GC [89]; MMP1 mRNA in ovarian cancer [90]; miR-25-3p, miR-130b-3p, miR-425-5p in colorectal cancer cells (CRC) [91]; miR-106b in lung cancer [92]; and miR-21 in oesophageal cancer [93]. Cell proliferation and invasion was induced by exosomal miR-1260b in lung adenocarcinoma [94] and miR-222 in PDAC [95]. Angiogenesis and tumor progression were influenced by exosome mediated Wnt4/β-catenin signaling in CRC [96] and by vascular endothelial growth factor A (VEGF-A) enriched exosomes in brain endothelial cells [97]. Tumor progression was augmented by exosomal miRNAs from TP53-mutant cells in colon cancer cells [98] and by exosomal lncRNA ZFAS1 in GC [92]. Exosomal miR-21 reduced apoptosis in GC cells [99], exosomal IL-6 induced metastasis in BC cells [100], exosomal HSP70 induced tumor progression in MSC cells [101], and exosomal TGF-β promoted tumor growth in LAMA84 cells [102]. Various recent studies based on the tumor promoting effect of exosomes have been listed in Table 2.

### 7.5. Exosomes and Drug Resistance

Exosomes form a physical barrier against drug penetration and confer drug resistance by transfer of cargoes from resistant to sensitive cells [104]. Exosome-mediated drug resistance may be devised through trafficking of non-coding RNAs, drug transporters and neutralization of antibody-based drugs, which has been described in the following sections.

#### 7.5.1. By Trafficking of Non-Coding RNAs

Non-coding RNAs, including miRNAs and lncRNAs, perpetuated drug resistance across an array of cancer cells. Exosomes from M2-macrophage exerted miR-21-mediated upregulation of PI3K/Akt signaling and reduced apoptosis and cisplatin resistance in GC [93]. Exosomal miR-221/222 modulated p27 and ERα for tamoxifen resistance [103] in BC cells. Exosomes derived from cisplatin resistant cells induced resistance in cisplatin sensitive A549 cells in a miR-100-5p-dependent manner [104]. In ovarian cancer cells, exosomal miR-443 induced senescence and resistance against paclitaxel [109]. In prostate cancer, CAF derived exosomes conferred gemcitabine resistance via Snail and miR-146a [105]. Exosomal cargo-lncRNA UCA1 mediated tamoxifen resistance [107] and lncRNA actin filament associated protein1 antisense RNA 1 (AFAP1-AS1) conferred trastuzumab resistance by binding to AU binding factor 1 and translating erythroblastic oncogene B2 (ERBB2) [106] in BC cells. MSC-derived exosomes aided the transfer of lncRNA PSMA3-AS1 to myeloma cells and exerted resistance against proteasome inhibitor [110]. In GC, exosomal lncRNA HoxA transcript at a distal tip (HOTTIP) made sensitive GC cells cisplatin resistant [108].

#### 7.5.2. By Trafficking of Drug Transporters and Neutralizing Antibody-Based Drugs

The exosome-mediated transfer of drug transporter molecules is intimately associated with the spread of drug resistance across diverse cancer forms. Exosomes transported P-glycoprotein (P-gp) from doxorubicin-resistant cells [68] and multidrug resistance protein-1 (MDR-1) from docetaxel-resistant cells [111] to confer drug resistance in sensitive BC cells. Recently, it has been evidenced that exosome-mediated transfer of chloride intracellular channel 1 upregulated P-gp and B cell lymphoma-2 (Bcl-2) and conferred vincristine resistance in GC cell line SGC-7901 [112].

B-cell lymphoma derived exosomes modulated ATP-binding cassette (ABC) transporter A3, carried CD20 antigen which shielded the cancer cells against therapeutic CD20 antibodies and evaded immune surveillance [113]. Exocytosis of TEXs from human epidermal growth factor receptor 2 (HER2) positive BC cells expressed specific decoy molecules and conferred resistance against monoclonal antibody trastuzumab, thus depicting that TEXs are also involved in neutralizing antibody based drugs [114].

## 8. Strategies against Tumor-Derived Exosomes

There have been, primarily, three approaches for the management of exosomes associated with pathogenesis, as described below.

### 8.1. Suppression of Exosome Biogenesis and Trafficking

Genetic knockdown of tumor suppressor TSG1 (protein involved with exosome biogenesis and trafficking) reduced Wnt5b-positive exosomes in colon cancer [115]. Suppression of annexin A1 (responsible for membrane contact sites, inward vesiculation and exosome biosynthesis) reduced the number of secreted exosomes in pancreatic cancer cells [116]. Manumycin A was reported to inhibit ESCRT-dependent exosome biogenesis by modulating Ras/Raf/ERK1/2/heterogeneous nuclear ribonucleoprotein H1 axis in prostate cancer cells [117].

Small molecule inhibitor GW4869 against nSMase2 reduced secretion of ceramide enriched exosomes [118] and sensitized breast tumors by inhibition of exosomal PD-L1 [119]. Knockout of nSMase2 reduced exosome secretion, angiogenesis and metastasis in breast tumors [120]. Another inhibitor of lipid metabolism, pantethine, a pantothenic acid (vitamin B_5_) derivative, depleted the release of exosomes in MCF-7 variants and increased doxorubicin responsiveness [121]. Genetic silencing of Rab27A/B reduced exosomal secretion by HNSCC and macrophages, thereby minimizing metastasis in BC cells [76] and lung metastasis in melanoma [122]. PRAS40 downregulated Akt, downstream of TGF-β, and mediated antagonistic effects against exosome secretion and chemoresistance in breast and lung cancer cells [123]. WEB2086, a platelet-activating factor receptor (PAFR) antagonist, was shown to reduce gemcitabine-induced exosome release in PAFR-positive pancreatic cancer cells [124]. Other exosome extrusion inhibitors, such as chloramidine, bisindolylmaleimide-I, imipramine, d-pantethine, and calpeptin, and calcium chelators, such as ethylene glycol bis (2-aminoethyl ether) tetra-acetic acid, increased responsiveness toward 5-FU in prostate and BC cells [125]. The inhibition of protease-activated receptor 2 by an anticoagulant, apixaban, which binds to the tissue factor–factor VIIa complex, downregulated the secretion of TF-bearing exosomes from pancreatic cancer cells [126]. Dasitinib inhibited exosome release and beclin-1/Vps34 mediated autophagy in imatinib resistant K562 cells [127]. Reduced exosome secretion by synthetic peptide (constructed with a derivative of the secretion modification region of HIV-1 Nef protein, a N-terminus anchored polyethylene glycol residue and a c-terminus cluster in peptide) [128] and by Docosahexaenoic acid (a polyunsaturated fatty acid) [34] inhibited metastasis and angiogenesis, respectively, in BC cells.

### 8.2. Depletion of Exosome Uptake

A synthetic nanoparticle, which is a prototype of high-density lipoprotein, was used as an agonist of the scavenger receptor type B-1 (SR-B1) which eliminated cholesterol from lipid rafts and prevented exosome uptake by SR-B1 expressing cancer cells [129]. Other agents, such as heparin sulfate proteoglycans, methyl-β cyclodextrin (molecule used for cholesterol removal from natural and artificial membranes) and dynasore (dynamin inhibitor), have been reported to abrogate exosome endocytosis in cancer cells [130]. Heparin and dynasore attenuated the uptake of multiple myeloma-derived exosome by bone marrow stromal cells and inhibited phosphorylation of STAT1, STAT3, and ERK1/2 signaling pathways [131]. Radiation-derived exosomes made the recipient cancer cells radiation-resistant and aggravated proliferation. Heparin and simvastatin attenuated radiation-derived exosome uptake by recipient cells in in vitro and in vivo models of glioblastoma [132].

### 8.3. Modulation of Harmful Exosomal Cargo and Inhibition of Exosome Dissemination

Alteration of exosomal cargoes was achieved by viral manipulation or by incorporation of viral proteins/RNA into secreted exosomes [133]. Curcumin culminated the immunosuppressive effect of exosomes in BC by deregulation of the ubiquitin-proteasome system and cargo sorting of ILVs [134]. Subscapular sinus CD169+ macrophages bound with exosomes restricted their interaction with B cells, promoting tumor progression [135]. Exosome release was inhibited by inhibitors like indometacin (COX2 inhibitor) in combination with rapamycin (interfere with MVB biogenesis) in B lymphoma cells, by suppressing ATP-binding cassette sub-family A member 3 expression of the lymphoma cells and induced the cells to undergo complement dependent cytolysis under the effect of drug rituximab [113].

### 8.4. Removal of Exosomes

A microfluidics-based technology-microscale acoustic standing wave technology facilitates clearance of exosomes from circulation [136]. Innate immune system in co-operation with opsonization effects of complement proteins may be used for elimination of exosomes [137]. Opsonization of exosomal markers CD9 and CD63 by targeting anti CD9 and anti CD63 antibodies elevated exosomes representation to the macrophages, leading to exosomes’ elimination, which suppressed lung metastasis in vivo [138]. In colorectal cancer, dimethyl amiloride depleted exosomes, thereby elevating cyclophosphamide efficacy against the cancer cells [139].

## 9. Cancer Management with Exosomes

Exosomes have emerged as a new arena of clinical interest due to their prospective use in diagnostic applications as potential biomarkers, for carrying specific information of their progenitor cells, as well as for being ideal candidates for liquid biopsy [56].

### 9.1. Preclinical Studies on Anticancer Potential of Exosomal Cargoes

Uptake of exosomal contents does not always confer procarcinogenic signaling. There are instances where exosomal proteins promoted anticarcinogenic signaling pathways, e.g., exosomal uptake with payload of gastrokine1 suppressed H-Ras/Raf/MEK/ERK-mediated gastric carcinogenesis in gastric epithelial cells [140]. The miR-375 carried by exosomes inhibited cell proliferation and invasive capability in colon cancer cells through Bcl-2 blocking [141]. Exosomal miR-520b derived from normal fibroblasts cells inhibited proliferation and migration of pancreatic cancer cells [142]. The migratory behavior of lung cancer cells was reduced by exosomal miR-497 through suppression of growth factors, cyclin E1 and VEGF [143]. Exosomal circulating RNA circ-0051443 inhibited tumor progression through apoptosis induction in HCC cells [144]. In BC cells, exosomal miR-100 derived from MSCs inhibited angiogenesis in vitro via modulating mTOR/HIF-1α/VEGF signaling [145].

### 9.2. Exosomes as Biomarkers

Cancer cells secrete exosomes ten times higher than normal cells, which makes TEXs major potential candidates for liquid biopsy needed for cancer diagnosis and prognosis [57]. The release of exosomes in the extracellular space also aids in cancer diagnosis by examining their increased levels in various body fluids, such as blood, ascites fluid, urine, and saliva [146]. Exosomal DNA represents the entire genome; therefore, liquid biopsies of plasma aid in early detection of cancer-specific mutations. Exosomal CD63 and caveolin-1 served as non-invasive markers of melanoma [121]. Exosomal lncRNA, either with miR-21 or alone, was correlated with tumor classification (III/IV), stage of tumor and lymph node/distant metastasis in many cancer types [5]. Differential expression of exosomal miR-150, miR-155, and miR-1246 in serum of normal individuals and acute myeloid leukemia patients detected minimal residual disease [147]. Phosphatidylserine present on the exosomal surface also serves as a biomarker for diagnosis of early-stage cancer [148]. However, exosomal biomarkers are often overshadowed by highly prevalent complex proteins of the body fluids. Exosome isolation from body fluids follows either of the three methods, namely differential centrifugation coupled with ultracentrifugation, immunoaffinity pull-down, and density gradient separation. Mining of exosomal biomarkers from body fluid of cancer patients has been explored with fluorescence-based analytical techniques, electrochemical aptamer-based detection methods, localized surface plasmon resonance and surface-enhanced Raman scattering [149]. Though exosome biomarker analysis has tremendous translational potential, a gold standard for exosome isolation under clinical settings is yet to be achieved [150]. Since there is no definite consensus for isolation of exosomes, the best suitable body fluid for exosome isolation is also under investigation.

### 9.3. Role of Exosomes in Immunotherapy and Vaccine Development

DCs and other antigen presenting cells (APCs) derived exosomes are loaded with specific drugs; miRNAs of interest or even exosomes alone are implemented to trigger immune response in the recipient individuals (Figure 4). DC-based exosomes, in therapy, are beneficial as they possess abundant surface lactadherin that helps in efficient exosome uptake [151]. The functional moieties, such as MHC-I, MHC-II, CD40, CD80, CD86 TNF, FasL, TRAIL and natural killer group 2D (NKG2D) ligands on the surface of DC-derived exosomes, facilitate in imparting innate and adaptive antitumor immune response [152]. DC-derived exosomes activated NK cells in NKG2D and interkeukin (IL)-15Rα ligand dependent mode, which restored 50% functionality of NK cells and was implemented as a cell free vaccination strategy [153]. The administration of adjuvants, such as IFN-γ, Toll-like receptor agonists, and polyinosinic: polycyctidylic acid, was explored for production of mature DC-derived exosomes which showed greater potential for activation of Th1 cells [154,155]. Immunogenic cell death was induced by melphalan, an anticancer drug, in multiple myeloma cells by increasing the damage-associated molecular pattern containing exosomes, thus triggering NK cell cytotoxicity [156]. A histone deacetylase inhibitor, MS-275, increased the release of Hsp70 and MHC-I polypeptide-related sequence B (MICB)-rich exosomes which induced NK cytotoxicity and lymphocyte proliferation [157]. Heat shock treatment increasing the immunostimulatory activities of TEXs has been demonstrated in A20 lymphoma/leukemia cells. Heat shock tumor derived exosomes were observed to possess more immune-stimulating activities due to elevated expression of MHC and increased levels of cytokines, such as IL-1β, IL-12p40, and TNF-α [158].

Exosomes have potential use in vaccine development because the surface-bound proteins on exosomes of APCs, DCs and tumor cells originate from the progenitor cell membranes [5]. Nanoscale immunotherapy treatments with TEX, DC-derived exosomes and ascitic cell-derived exosomes have shown efficacy in stimulating the body’s immune system against cancer cells [159]. Ascitic cell-derived exosomes obtained from peritoneal cavity fluid of cancer patients triggered cancer cell lysis via activation of dendritic cells and MHC-1-dependent T cell response. Membrane-bound Hsp70 of TEX exhibited robust priming of T helper cell 1 (Th1)- and NK-mediated antitumor immune response [160]. Chemotherapy accompanied with hyperthermia has evolved as a new treatment mode for cancer involving TEXs. For instance, heat stress has increased the antitumor effect of TEXs derived from doxorubicin-treated MCF-7 cells [161]. DC-derived exosomes control tumor growth by eliciting CD8+ and CD4+ T cell responses [162]. DC-derived exosomes incubated with cancer antigen triggered cancer specific T cell response [163]. Adjuvant-based exosomal vaccines are effective in eliciting immune response. For example, streptavidin-lactadherin protein fused with immunostimulatory biotinylated CpG DNA (adjuvant) after transfection into murine melanoma cells created genetically modified exosomes. These exosomes have the ability to trigger improved antigen presentation to the DCs and other immune cells, contributing to enhanced immune response [164]. DC-derived exosomes have been observed to be more efficient as cell-free vaccines in treating malignancies that respond poorly to immunotherapy. For instance, α-feto protein-rich DC derived exosomes triggered more effective antitumor immune responses and modulated the TME in a HCC mice model [165]. Recently, it was observed that vaccination with TEX-pulsed DC along with cytotoxic drugs specifically targeted immunosuppressive MDSCs in pancreatic cancer cells [166]. DNA vaccines prepared by fusing ovalbumin antigen with lactadherin present on exosomal surface diminished fibrosarcoma, thymoma and melanoma metastasis by activating T lymphocytes [167].

### 9.4. Exosome-Based RNA Therapy

Exosome-based miRNA therapy exhibited immunosuppressive properties by controlling the gene expressions [19]. An early study reported that exosomes derived from human embryonic kidney cells were effective in regressing tumor growth by delivering miR-let7a in an EGFR-positive BC xenograft model [168]. The MSCs transfected with miR-124a enhanced exosomes carrying the RNA of interest production, which, when implemented against gliomas, reduced the cell viability and targeted FOXA2 that caused accumulation of lipids [169]. Transfer of lncRNA PTEN pseudogene 1 by exosomes derived from normal cells to bladder cancer cells reduced tumor progression in vitro and in vivo [170].

Exosomes also mediated targeted delivery of siRNA, e.g., siRNA transfected into exosomes targeted RAD51 and RAD52 in Hela and fibrosarcoma cells, which inhibited proliferation of the recipient cells [171]. Engineered exosomes containing IL-3 ligand or functional siRNA for BCR-ABL were successfully used against imatinib resistance in chronic myeloid leukemia patients [172]. Exosomes used for trafficking RNA interference (RNAi) mediators counteracted against oncogenic KRAS and improved overall survival in mouse models of pancreatic cancer [173]. Delivery of engineered exosome mediated siRNA inhibited post-operative metastasis of BC, indicating a promising strategy against tumor progression [174]. Successful delivery of antisense miRNA oligonucleotides against miR-21 by electroporating them in exosomal membrane improved the treatment efficacy for glioblastoma by inducing the expression of PTEN and PDCD4, resulting in decreased tumor size [175].

### 9.5. Exosomes in Stem Cell Therapy

Normal stem cell-derived exosomes are free of tumorigenic factors and are potential candidates for stem cell therapy [176]. MSC-derived exosomes can protect their cargoes from degradation, facilitate easier uptake by recipient cells, elicit low toxicity and immunogenicity, and these exosomes can be modified to enhance cell type-specific targeting and may be a prospective tool for cell-free based therapeutic approaches [177]. Exosomal miR-144 derived from bone marrow derived MSC retarded the spread of NSCLC by targeting cyclin E1 or E2 [178]. Exosomes released from miR-101-3p overexpressing MSCs negatively affected the proliferation and migration of oral cancer cells by targeting the collagen type X α1 chain [179]. MSC-derived exosomes were genetically engineered by loading them with polo-like kinase 1 (PLK-1)-siRNA and were utilized for PLK1 gene silencing in bladder cancer [180]. The primary hurdles of stem cell-based therapy, such as teratoma formation and embolization, are less frequent with exosome-based stem cell therapeutics. Exosomes secreted from induced pluripotent stem cells may exert better therapeutic effects [163].

### 9.6. Exosomes in Drug Delivery

Normal cell derived exosomes exhibit excellent biodistribution, biocompatibility, low immunogenicity, capacity to cross the blood–brain barrier and high target specificity, which make them potential candidates for drug delivery in cancer [5]. The exosomal surface proteins regulate efficient drug delivery because of their involvement in exosomes uptake by the tumorigenic recipient cells [181]. Exosomes derived from androgen-sensitive human prostate adenocarcinoma cells carrying paclitaxel negatively affect the cancer cells’ viability [182]. DC-derived exosomes in BC and macrophage-derived exosomes in lung cancer were loaded with the drugs trastuzumab and paclitaxel, respectively, and successfully delivered to the recipients [183,184]. Moreover, exosomes loaded with doxorubicin conjugated with gold nanoparticles showed anticancer effect against lung cancer cells [185]. Exosomes with A disintegrin and metalloproteinase 15 (ADAM15) expression (A15-Exo) co-delivered with doxorubicin and cholesterol-modified miRNA 159 exhibited anticancer effect in BC cells [186]. Paclitaxel loaded exosomes showed sensitivity towards MDR cancer cells via by-passing P-gp-mediated drug efflux and also inhibited metastasis in a lung cancer xenograft model [187]. Unmodified exosomes encapsulated with doxorubicin reduced tumor proliferation in a mouse mammary carcinoma xenograft model [137]. Exosomal delivery of doxorubicin induced its therapeutic activity in xenograft models of breast and ovarian cancer [188]. Exosomes isolated from engineered immature DCs (expressed Lamp2b fused with αv integrin-specific iRGD peptide (CRGDKGPDC)) loaded with doxorubicin successfully targeted αv integrin-positive breast tumor cells [189]. Exosome encapsulated gemcitabine exhibited anticancer properties in autologous pancreatic cancer cells and in a xenograft model [190].

Phytochemicals, administered via an exosome-mediated drug delivery system, can provide health benefits and anticancer properties [56]. Pancreatic adenocarcinoma cell-derived exosomes aided curcumin in inflicting its anticancer properties among tumor cells [191]. Milk-derived exosomes encapsulated with anthocyanidins exhibited antiproliferative effect in a xenograft lung carcinoma model [192]. Exosomal formulations of black bean extract exhibited pronounced antiproliferative effect in many cancer cells [193]. Exosomal formulations with berry anthocyanidins exhibited anticancer properties in ovarian cancer with enhanced sensitivity in chemoresistant tumors [194]. Exosomal encapsulation of celastrol (a triterpenoid) exhibited antiproliferative effect in lung cancer cells and in a xenograft model [195]. Recent studies on exosomal drug delivery of chemotherapeutic drugs and phytochemicals are listed in Table 3.

### 9.7. Induction of Chemosensitivity with Exosomes

TEXs impart drug resistance but may also be used for inducing drug sensitivity. Dimethyl amiloride augmented ABC transporter containing exosome secretion revived the cyclophosphamide sensitivity of cancer cells [31]. Downregulation of the GAIP-interacting protein C terminus mediated secretion of ABCG2 drug transporters containing exosomes and suppressed gemicitabine resistance in pancreatic cancer cells [196]. In oral squamous cell carcinoma, exosomal miR-155 increased chemoresistivity in cisplatin-sensitive cancer cells [197]. The exosomes loaded with CRISPR/Cas9 induced apoptosis and cisplatin chemosensitivity in ovarian cancer cells [198]. An increase in apoptosis and chemosensitivity was observed in cisplatin-resistant human gastric adenocarcinoma cells through treatment with si-c-Met containing exosomes derived from human kidney epithelial cell line [199]. Normal intestinal FHC cell-derived exosomes transferred miR-128-3p into oxiplatin resistant CRC cells which induced their chemosensitivity and decreased motility [200]. miR-122-transfected adipose tissue-derived MSCs (AMSCs) released exosomes carrying miR-122 and, when cocultured with hepatocyte carcinoma cells, induced sorafenib chemosensitivity [201]. miR-567 induced chemosensitivity in resistant BC cells towards trastuzumab and blocked autophagy [202]. Exosomal miR-200c induced chemosensitivity towards docetaxel and apoptosis in tongue squamous cell carcinoma [203]. Coculture of miR-199a carrying exosomes derived from AMSCs with HCC cells downregulated mammalian target of rapamycin (mTOR) pathway and induced chemosensitivity towards doxorubicin [204]. Various recent reports on exosome-mediated reversal of chemosensitivity have been listed in Table 4.

### 9.8. Exosomes in Clinical Trials

According to the National Institutes of Health website, a large number of clinical trials are being conducted with exosomes (Table 5). In a study, plant exosomes were modified to deliver curcumin in colon cancer patients (ClinicalTrials.gov Identifier: NCT01294072). Phase I and II clinical trials with DC-derived exosomes indicated activation of T cell- and NK cell-based immune responses in NSCLC patients [154]. A phase II clinical trial (ClinicalTrials.gov Identifier: NCT01159288) on NSCLC observed that exosomes derived from TLR4L-or interferon-γ (IFN-γ)-maturated DCs enriched with MHC I- and MHC II-restricted cancer antigens as maintenance immunotherapy subsequent to first-line chemotherapy [205]. A study on HER2-positive BC patients measured HER2-HER3 dimer expression in exosomes (ClinicalTrials.gov Identifier: NCT04288141). Another trial led to a therapeutic analysis on cancer-derived exosomes via treatments with lovastatin and vildagliptin in thyroid cancer patients (ClinicalTrials.gov Identifier: NCT02862470). Characterization of exosomal non-coding RNAs was carried out in cholangiocarcinoma patients (ClinicalTrials.gov Identifier: NCT03102268). Another study reported exosome-mediated intercellular signaling in pancreatic cancer (ClinicalTrials.gov Identifier: NCT02393703). In metastatic pancreatic adenocarcinoma, exosomes with KrasG12D siRNA were used to treat pancreatic cancer with KrasG12D mutation (ClinicalTrials.gov Identifier: NCT03608631). In head and neck cancer, the effects of metformin hydrochloride on cytokines and exosomes were investigated (ClinicalTrials.gov Identifier: NCT03109873). A phase I clinical trial (ClinicalTrials.gov Identifier: NCT01668849) investigated the ability of plant exosomes to prevent oral mucositis induced by combined chemotherapy and radiation in head and neck cancer patients. However, more clinical trials are needed with modified exosomes which may exhibit anticancer effect.

## 10. Current Limitations and Challenges

Exosomes mediate intercellular communication and play significant roles in both physiological and pathological processes. A new hypothesis suggested that the target cells inhibit the incoming signals by forming exosome dimers based on the particle size, zeta potential and/or ligand–receptor pairs which facilitates cancer metastasis, cancer immunoregulation, intraocular pressure homoeostasis, tissue regeneration and many others [206].

Exosomes released by normal and malignant cells are endowed with heterogeneity and pleiotropic physiological and pathological effects. Inhibition of the release of TEXs may have both anti-carcinogenic and pro-carcinogenic effects. The majority of the exosome released inhibitors are not cancer-specific and also affect normal cells. Therefore, inhibition of exosome release may act as a double-edged sword which should be carefully manipulated for minimal adverse effects [34].

Isolation of pure and specific exosomes is limited by technical constraints, the availability of suitable biomarkers for specific exosomes, and expensive technologies [5]. A major hurdle in the execution of liquid biopsy is isolation of exosomes by an economic user-friendly tool. Protein contaminated and heterogenous exosome pool is obtained using ultracentrifugation. Asymmetric flow field-flow fractionation, though a prospective tool, needs technical expertise and requires a huge amount of initial sample. Other exosome isolation methods like microfluidic devices, sucrose gradients, size exclusion chromatography, and affinity-based exosome isolation kits are accompanied with both advantages and disadvantages like lack of robustness and specificity [31]. A perfect exosome isolation method should be robust, reproducible, specific, economic and user friendly as a diagnostic tool.

Detailed research of exosome biogenesis, functional diversity of exosomes and the identification of cancer specific biomarkers may be effective for exosome-based therapeutic approaches with minimum adverse effects [34]. Determination of exosomal cargo sorting and releasing mechanisms holds great potential for the development of various applications in cancer research [31].

Normally, less than 1 μg of exosomal protein is yielded from 1 mL of culture medium, whereas the majority of studies have reported 10–100 μg of exosomal protein as an effective dose for in vivo models [163]. The introduction of exosome-mimetic vesicles (100–200 nm in diameter) has conquered exosomal limitations like low loading efficiency and low yields. These nanovesicles have been used for delivery of chemotherapeutic drugs [204,205] and RNAi [207] to target cancer cells. Hybrid nanocarriers formed by the fusion of exosomes with liposomes changed the exogenous lipid composition and was effective in the delivery of chemotherapeutic drugs [208].

## 11. Conclusions

It may be deciphered that the intercellular communication via exosomes is evident throughout cancer progression. Apart from cancer pathogenesis, exosome biology heralds the future arena of non-invasive diagnostic tools for cancer management, especially in the spheres of liquid biopsy, immunotherapy and vaccine development, RNA therapy, stem cell therapy, drug delivery, and reversal of chemoresistance. Preclinical studies have undoubtedly proven the immense potential of exosomes in cancer therapeutics, but a number of clinical trials have failed to achieve this success. These inconsistent results indicate major challenges including in-depth knowledge of exosome biogenesis and protein sorting, perfect and pure isolation of exosomes, large scale production, better loading efficiency targeted delivery of exosomes. These hurdles have to be confronted before successful implementation of exosomes for the diagnosis and therapy of cancer. This review has attempted to envisage the implication of exosomes in cancer pathogenesis and cancer therapeutics along with the current limitations so that researchers may be made aware of the existing lacunae with regard to exosomes in their use against cancer. This knowledge may help scientists to improvise innovative technologies for successful translation of the exosome-mediated diagnosis and treatment of malignant neoplasms.

## Figures and Tables

**Figure 1 cancers-13-00326-f001:**
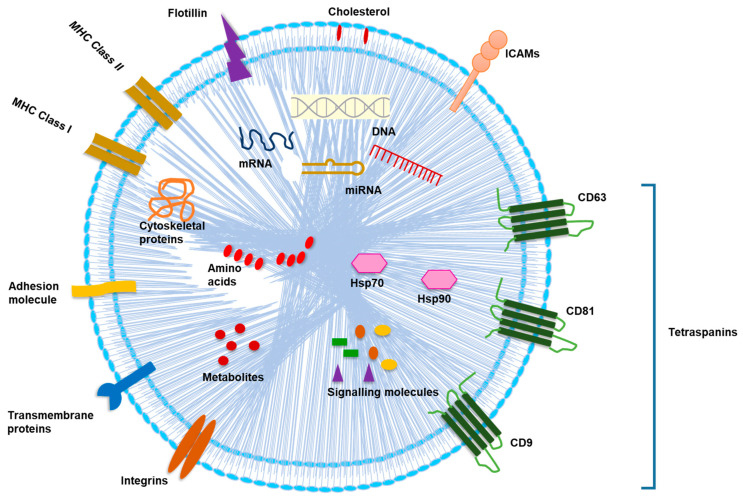
Structure of exosome with membrane proteins and cargoes. Exosomes consist of many constituents of a cell including DNA, RNAs, amino acids, proteins, metabolites, enzymes, lipids (cholesterol) and Hsps along with several cytosolic and cell-surface signaling proteins which are involved in intercellular communications. Exosomal membrane is rich in transmembrane proteins (tetraspanins such as CD81, CD63 and CD9), flotillin, ICAMs, integrins and adhesion molecules. They consist of immune components including MHC class I and class II molecules. Abbreviations: CD, cluster of differentiation; DNA, deoxyribonucleic acids; Hsps, heat shock proteins; ICAMs, intercellular adhesion molecules; MHC, major histocompatibility complex; mRNA, messenger RNA; miRNA, microRNA.

**Figure 2 cancers-13-00326-f002:**
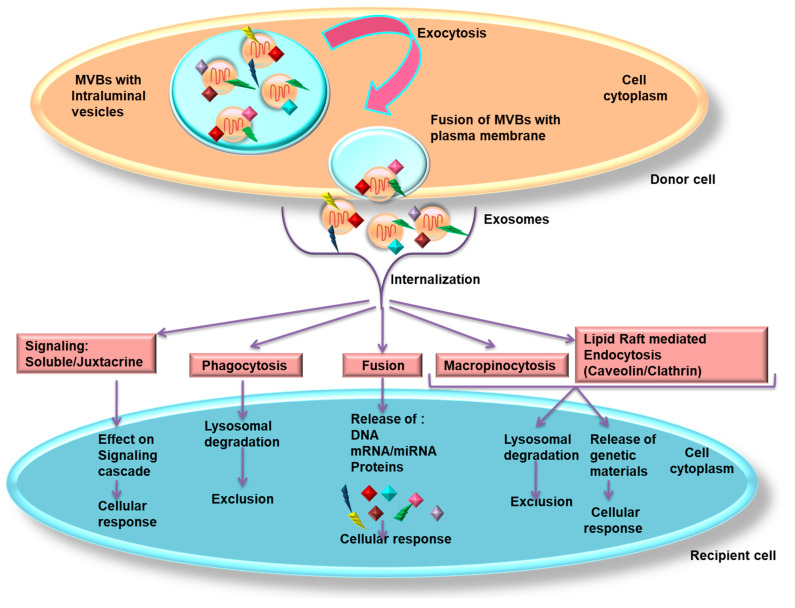
Mechanisms of internalization of exosomes. The exosomes inside the MVBs are extruded out from the donor cells by exocytosis on merging with plasma membrane. The released exosomes are then internalized via different modes: soluble/juxtacrine signaling; phagocytosis; fusion; micropinocytosis and lipid raft mediated endocytosis. The lipid raft mediated endocytosis can be either clathrin or caveolin protein dependent. Exosomes internalized by soluble/juxtacrine signaling affect the signaling cascade of the recipient cell. During phagocytosis, the exosomes undergo degradation, whereas, in the fusion event, genetic material is released that causes cellular response. In macropinocytosis and lipid raft-mediated endocytosis, the exosomes either undergo lysosomal degradation or mediate cellular response. Abbreviations: mRNA, messenger RNA; miRNA, microRNA; MVBs, multivesicular bodies.

**Figure 3 cancers-13-00326-f003:**
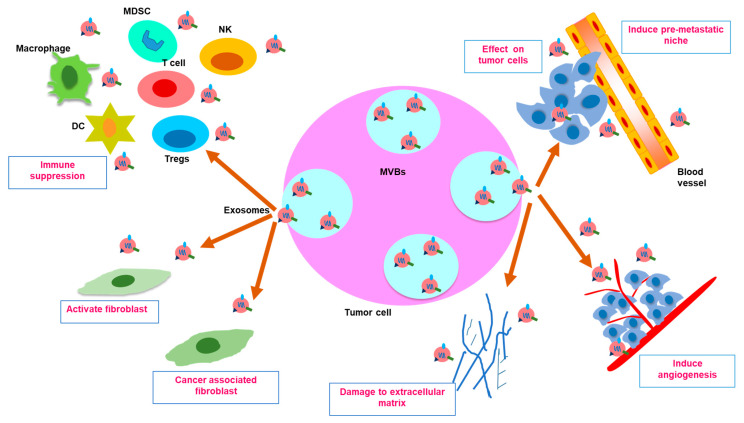
Exosomes in tumor microenvironment. Exosomes secreted from tumor cells containing MVBs exhibited a dynamic signaling between tumor cells and the TME. Exosomes may lead to immune suppression by downregulating macrophages, DC, T cells and NK cells and upregulating immunosuppressive cells like Tregs, MDSCs and TAMs. Exosomes induced differentiation of fibroblasts, activation of CAFs and degradation of ECM, which are associated with TME construction. They are involved in the alteration of ECM, hypoxia-mediated angiogenesis and the formation of pre-metastatic niches that trigger the metastatic escape of tumor cells. Abbreviations: CAFs, cancer-associated fibroblasts; ECM, extracellular matrix; DCs, dendritic cells; MDSCs, myeloid-derived suppressor cells; NK cells, natural killer cells; TAMs, tumor-associated macrophages; TME, tumor microenvironment; Tregs, tumor regulatory cells.

**Figure 4 cancers-13-00326-f004:**
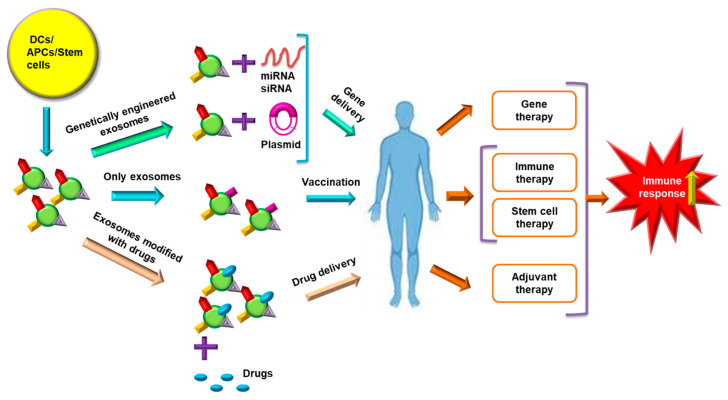
Exosomes in therapeutic approaches. Exosomes derived from DCs, APCs and stem cells can be utilized for immunotherapy, gene therapy, stem cell therapy and adjuvant therapy. Exosome based gene therapy is obtained by genetically engineered exosomes loaded with miRNA, siRNA and plasmids of interest. Stem cell or DC-derived exosomes can be implemented alone as vaccines and confer stem cell-based therapy or immunotherapy. The exosomes can also be utilized for drug delivery vectors by modifying them with drugs of interest. The DC-, APC- and stem cell-derived exosomes administered into the patient help in triggering immune response in combating cancer by targeting and regulating the mechanisms against which the exosomes are implemented. Abbreviations: APCs, antigen presenting cells; DCs, dendritic cells; miRNA, microRNA; siRNA, small interfering RNA.

**Table 1 cancers-13-00326-t001:** Different types of Rabs and their function in endocytic trafficking.

Rabs	Effects	Functions	References
Rab27	Secretion of exosomes	Release of markers MHC II, CD63, and CD81 in cancer cells	[32]
Rab7, Rab27a/b		Fusion with plasma membrane	[43]
Rab5, Rab4, Rab35	Recycling	Fast delivery of cargo to the plasma membrane	[36]
Rab5, Rab11a, Rab11b, Rab25		Slow delivery of cargo to the plasma membrane	[37,38]
Rab9		Transportation to TGN	[39]
Rab5, Rab7	Endosome maturation	Release of Rab5	[40]
Rab7	Sorting and degradation	Reduction in pH and acquisition of hydrolytic enzymes	[41]
Rab5 overexpression<break/>Note: may be rescued by Rab7	Suppression of release of exosomal markers syndecan, CD63, and ALIX	Inhibition of progression of endocytosed material from early endosomes	[44]

Abbreviations: ALIX, ALG-2 interacting protein X; MHC-II, major histocompatibility complex II; TGN, trans-Golgi network.

**Table 2 cancers-13-00326-t002:** Tumor-promoting effects of exosomal cargoes on recipient cells.

Exosome Donor Cells	Exosomal Cargo	Target Cells	Effects	Mechanisms	References
Human prostate cancer (PC3) cells	Integrin α_V_β_6_	Peripheral blood mononuclear cells and THP-1 monocyte cells	↑M2 polarization	↓STAT1/MX1/2 signaling	[73]
		Human prostate cancer DU145 cells	↑Cell adhesion and migration	↑Latency-associated peptide-TGF-β	[84]
HCC (mouse Hepa1-6, H22, and human HepG2, H7402) cells	miR-146a-5p	Mouse RAW264.7 cells, THP-1 cells, mice peritoneal macrophages	↑Pro-inflammatory factors, ↑M2 polarization, ↑T-cell exhaustion by M2 macrophages	↑NF-κB, ↑p-STAT3, ↓p-STAT1	[74]
Human Bladder cancer (T24 and 5637) cells	miR-663b	T24 and 5637 cells	↑Cell proliferation, ↑EMT	↓ERF, ↓E-cadherin, ↑Vimentin	[81]
Human PDAC (Hs 766 T) and metastatic (Hs 766T-L2) cells	lncRNA-Sox2ot	Human PDAC (BxPC-3) cells	↑EMT, ↑stemness, ↑invasion and metastasis	↑Sox-2	[82]
Human bladder cancer (T24) cells	miR-21	Human THP-1 cell-derived macrophages	↑M2 polarization, ↑tumor cell migration and invasion	↓PTEN, ↑PI3K/Akt-STAT3 signaling	[85]
M2 polarized macrophages (TAMs)	Apolipoprotein E	Mouse gastric carcinoma (MFC) cells	↑Cell migration	↑PI3K-Akt signaling	[86]
Human NSCLC (A549 and H1299) cells	lncRNA UFC1	A549 and H1299 cells	↑Cell proliferation, ↑migration, ↑invasion	↓PTEN via EZH2-mediated epigenetic silencing	[87]
Human GC (BGC-823) cells	lncRNA-ZFAS1	Human GC (MKN-28) cells	↑EMT, ↑cell proliferation, ↑migration	↑Cyclin D1, ↑Bcl-2, ↓Bax, ↓E-cad, ↑N-cad, ↑Slug	[88]
Human GC (SGC7901) cells	EGFR	Primary mouse liver cells	↑Cell proliferation, ↑metastasis	↓miR-26a/b, ↑HGF, ↑c-Met	[89]
Human CRC (HCT116) cells	miR-25-3p, miR-130b-3p and miR-425-5p	Macrophages RAW264.7	↑M2 polarization, ↑EMT, ↑liver metastasis	↑CXCL12/CXCR4 axis, ↓PTEN, ↑PI3K-Akt signaling	[91]
Human lung cancer (SPC-A-1 and H1299) cells	miR-106b	SPC-A-1 and H1229 cells	↑Migration and invasion	↓PTEN	[92]
Human esophageal cancer (EC9706) cells	miR-21	EC9706 cells	↑Metastasis	↓PDCD4, ↑MMP2, ↑MMP9	[93]
Human lung adenocarcinoma (H1299) cells	miR-1260b	Human A549 cells	↑Cell invasion, ↑cell proliferation, ↑drug resistance	↑Wnt/β-catenin signaling, ↓sFRP1, ↓Smad4	[94]
Human PDAC (Capan-1 and Hs 766T-L3) cells	miR-222	PDAC (Capan-1 and Hs 766T-L3 cells)	↑Cell invasion, ↑metastasis	↑Akt, ↓PPP2R2A, ↑p-P27	[95]
Hypoxic human CRC (HT29 and HCT116) cells	Wnt4	Endothelial (HUVECs) and CRC (HT29) cells	↑Proliferation, ↑angiogenesis, ↑migration	↑β-Catenin signaling	[96]
TP53-mutant (HT29) colon cancer cells	miR-1249-5p, miR-6737-5p, and miR-6819-5p	Human colon fibroblasts (CCD-18Co) cells	↑Tumor progression	↓TP53	[98]
Murine bone marrow–derived macrophages	miR-21	Human GC (MFC, MGC-803) cells	↓Apoptosis, ↑resistance to cisplatin	↑PI3K/AKT signalling, ↓PTEN	[99]
Co-culture of THP-1-derived macrophages exposed to apoptotic human BC (MCF-7 or MDA-MB-231) cells	IL-6	Naive (MCF-7 or MDA-MB-231) cells	↑Proliferation, ↑metastasis	↑p-STAT3, ↑cyclin D1, ↑MMP2, ↑MMP9	[100]
Human lung cancer (A549) cells	HSP70	MSCs extracted from human adipose tissue	Pro-inflammatory MSCs, ↑tumor growth	↑TLR-2/NF-κB signaling, ↑IL-6, ↑IL-8, ↑MCP-1	[101]
Human chronic myeloid leukemia (LAMA84) cells	TGF-β	LAMA84 cells	↑Proliferation, ↓apoptosis, ↑tumor growth	↑SMAD 2/3, ↑Bcl-w, ↑Bcl-xL, ↑survivin, ↓BAD, ↓BAX, ↓PUMA	[102]
Human BC (MCF-7) tamoxifen resistant cells	miR-221/222	Human BC (MCF-7) wild type cells	↑Resistance to tamoxifen	↓P27, ↓ERα,	[103]
Human cisplatin resistant A549 cells	miR-100-5p	Human A549 cells	↑Resistance to cisplatin	↑mTOR	[104]
Gemcitabine treated human PDAC CAFs	Snail and miR-146a	Human pancreatic cancer L3.6pl cells	↑proliferation, ↑resistance to gemcitabine	↑Snail, ↑miR-146a	[105]
Human HER-2-positive BC trastuzumab resistant (SKBR-3 and BT474) cells	lncRNA AFAP1-AS1	SKBR-3 and BT474 cells	↑Resistance to trastuzumab	↑ERBB2	[106]
Tamoxifen resistant BC (LCC2) cells	lncRNA UCA1	ER-positive BC MCF-7 cells	↑Cell viability, ↑resistance to tamoxifen	↓caspase-3	[107]
Human GC (MGC-803 and MKN-45) cisplatin resistant cells	lncRNA HOTTIP	MGC-803 and MKN-45 cells	↑Resistance to cisplatin	↑HMGA1	[108]

Symbols: ↑, upregulated; ↓, downregulated; Abbreviations: AFAP1-AS1, actin filament associated protein1 antisense RNA 1; Akt, protein kinase B; Bad, Bcl-2 associated agonist of cell death; Bax, Bcl-2-associated X protein; Bcl-2, B-cell lymphoma 2; c-Met, Mesenchymal-epithelial transition factor; CXCL12, C-X-C motif chemokine ligand 12; CXCR4, C-X-C chemokine receptor type 4; Erα, estrogen receptor-α; ERF, Ets2-repressor factor; ERBB2, erythroblastic oncogene B; HGF, hepatocyte growth factor; HMGA1, High-mobility group A1; HOTTIP, HOXA transcript at the distal tip; MCP-1, monocyte chemoattractant protein-1; MMP, matrix metalloproteinase; NF-κB, nuclear factor kappa-light-chain-enhancer of activated B cells; PDCD4, programmed cell death 4; PI3K, phosphoinositide 3-kinase; PPP2R2A, protein phosphatase 2 regulatory subunit B alpha; PTEN, phosphatase and tensin homolog; PUMA, p53 upregulated modulator of apoptosis; sFRP, secreted frizzled-related protein 1; STAT, signal transducer and activator of transcription; Sox-2, sex determining region Y-box 2; TGF-β, transforming growth factor-β; TLR-2, toll-like receptor 2; TP53, tumor protein p53.

**Table 3 cancers-13-00326-t003:** Exosomes as delivery system for therapeutic implications against cancer.

Exosome Source	Modification of Exosomes with Drugs	Loading Method	Target Cells	Effect	Mechanism	References
Chemotherapeutic drugs						
Human mammary adenocarcinoma cells (M-CF-7), mouse mammary carcinoma cells (4T1), and human prostate adenocarcinoma cells (PC3)	Doxorubicin	Incubation	4T1 tumor-bearing BALB/c mice	↓Tumor growth, but no significant reduction in tumor growth with exosomes loaded with doxorubicin compared to free drug	-	[137]
Human prostate cancer (LNCaP and PC-3) cells	Paclitaxel	Incubation	LNCaP and PC-3 cells	↑Cytotoxic effect of paclitaxel	-	[182]
Human NSCLC (H1299) cells	Exo-gold nanoparticles-doxorubicin	Incubation	Human NSCLC (H1299 and A549) cells	↑DNA damage, ↑apoptosis	↑caspase-9, ↑ROS	[185]
Human monocyte (THP-1 cells)-derived macrophages	A15-Exo-doxorubicin-cho-miR159	Mixing in triethylamine solution overnight, Incubation	αvβ3+ and αvβ3- human BC (MDA-MB-231 and MCF-7) cells	↓Cell proliferation, ↑apoptosis	↓TCF7, ↓MYC	[186]
			MDA-MB-231 tumor-bearing BALB/c-nu mice	↓Tumor growth, ↑survival of mice	↓TCF7, ↓MYC, ↓Ki67, ↓CD31	
Mouse immature dendritic cells (imDCs)	Doxorubicin	Electroporation	MDA-MB-231 tumor-bearing BALB/c nude mice	↓Tumor growth	-	[189]
Human pancreatic cancer (Panc-1) cells	Gemcitabine	Sonication	Panc-1 and A549 cells	↓Cell viability	-	[190]
			Panc-1 tumor-bearing BALB/c nude mice	↓Tumor volume	↓Alanine aminotransferase, ↓aspartate aminotransferase, ↓TNF-α, ↓IL-6 in exo-gemcitabine group compared to gemcitabine	
Mouse (RAW 264.7) macrophages	Paclitaxel	Sonication	Murine Lewis lung carcinoma cell subline (3LL-M27 cells), Madin-Darby canine kidney (MDCK-WT) and (MDCK-MDR1) cells	↑Drug cytotoxicity, ↑chemosensitization of MDR cells	-	[187]
			8FlmC-FLuc-3LL-M27 tumor bearing C57BL/6 mice	↓Metastasis	-	
Human BC (MDA-MB-231) cells and mouse ovarian cancer (STOSE) cell	Doxorubicin	Electroporation	MDA-MB-231 and STOSE tumor bearing FVB/N mice	↑Doxorubicin efficacy, ↓tumor volume	-	[188]
Phytochemicals						
Human pancreatic adenocarcinoma (PANC-1, MIA PaCa-2) cells	Curcumin	Incubation	PANC-1 and MIA PaCa-2 cells	↓Cell viability,	-	[191]
Pooled raw milk from Jersey cows	Anthocyanidins	By mixing in a solution of acetonitrile: ethanol (1:1 *v*/*v*) and PBS	Human pancreatic cancer (PANC1 and Mia PaCa2), lung cancer (A549 and H1299), colon cancer (HCT116), BC (MDA-MB-231 and MCF7), prostate cancer (PC3 and DU145), and ovarian cancer (OVCA432) cells	↓Cell proliferation, ↓cell survival	↓NF-κB	[192]
			A549 tumor bearing female athymic nude (nu/nu) mice	↓Tumor growth	--	
MCF7, PC3, human liver (HepG2), colon cancer (Caco2) cells	Black bean extract	Electroporation	MCF7, PC3, HepG2 and Caco2 cells	↓Cell viability	--	[193]
Mature bovine milk	Anthocyanidins	By mixing	Human ovarian cancer (A2780, A2780/CP70, OVCA432, and OVCA433) cells	↓Cell survival	-	[194]
			A2780 tumor-bearing female athymic nude mice	↓Tumor volume	-	
Milk from pasture-fed Holstein and Jersey cows	Celastrol	By mixing	Human lung cancer (H1299 and A549) cells	↓Cell survival,	-	[195]
			H1299 and A549 tumor-bearing female athymic nude mice	↓Tumor volume	-	

Symbols: ↑, upregulated; ↓, downregulated; Abbreviations: MYC, master regulator of cell cycle entry and proliferative metabolism; NF-κB, nuclear factor kappa-light-chain-enhancer of activated B cells; ROS, reactive oxygen species; TCF7, transcription factor 7; TNF-α, tumor necrosis factor-α.

**Table 4 cancers-13-00326-t004:** Reversal of chemoresistance in resistant cancer cells with exosomal cargoes.

Exosome Source	Modification in Exosomal Cargo Content	Target Cells	Effects	Mechanisms	References
Human mesenchymal stem cells (MSCs)	Anti-miR-9	Glioblastoma (U87 and T98G) cells	↑Apoptosis ↑chemosensitivity towards temozolomide	↑Caspase-3 ↓P-gp ↓MDR1	[151]
Human kidney epithelial (HEK293T) cells	si-c-Met	Human gastric adenocarcinoma (SGC7901and SGC7901/DDP) cells	↑Apoptosis ↑chemosensitivity towards cisplatin	↓c-Met gene	[199]
Normal intestinal foetal human cells (FHC)	miR-128-3p	Human oxiplatin resistant colorectal cancer (HCT116OxR) cells	↑Oxiplatin accumulation ↑apoptosis ↓proliferation ↓self-renewal	↓Bmi1 ↓MRP5 ↓N-cadherin ↑E-cadherin	[200]
Human adipose tissue derived mesenchymal stem cells (AMSCs)	miR-122	Human HCC (HepG2, Huh7) cells	↑Apoptosis ↑cell cycle arrest ↑chemosensitivity towards sorafenib	↑G0/G1 arrest ↓CCNG1 ↓ADAM10 ↑Caspase-3 ↑Bax	[201]
Human normal breast epithelial (MCF 10A) cells	miR-567	Human trastuzumab resistant BC (SKBR-3/TR and BT474/TR) cells	↑Chemosensitivity towards trastuzumab ↑autophagy	↓ATG5 ↑p62 ↓LC3-11	[202]
Human normal tongue epithelial (NTECs) cells	miR-200c	Docetaxel resistant hepatic stellate cells (HSC-3DR) cells	↑Chemosensitivity towards docetaxel ↑apoptosis	↓TUBB3 ↓PPP2R1B	[203]
Human adipose tissue derived mesenchymal stem cells (AMSCs)	miR-199a	Human colorectal cancer (CRC) (Huh7, SMMC-7721, PLC/PRF/5) cells	↑Chemosensitivity towards doxorubicin	↓mTOR	[204]

Symbols: ↑, upregulated; ↓, downregulated; Abbreviations: ADAM10, A disintegrin and metalloproteinase 10; ATG5, autophagy related 5 protein; Bax, Bcl-2-associated X protein; BC, breast cancer; c-MET, mesenchymal epithelial transition factor; CCNG1, Cyclin G1; LC3, microtubule associated protein PIA/IB-light chain 3-I; MDR1, multidrug resistance protein-1; MRP5, multidrug resistant associated protein 5; mTOR, mammalian target of rapamycin; P-gp, P-glycoprotein; PPP2R1B, protein phosphatase 2 scaffold subunit 1β; TUBB3, class III β-tubulin gene.

**Table 5 cancers-13-00326-t005:** Clinical trials on exosomes.

Trial No. (ClinicalTrials.gov Identifier:)	Study Type	Cancer Type	Study Perspective	Study Design	Status
NCT01294072	Phase I	Colon cancer	Interventional	Investigation of the ability of plant-derived exosomes to deliver curcumin	Active, not recruiting
NCT01159288	Phase II	Non-small cell lung cancer	Interventional	Trial of a vaccination with exosomes derived from dendritic cell loaded with tumor antigen	Completed
NCT04288141	Observational	Early HER2-positive BC, Metastatic HER2-positive BC	Prospective	Assessment of HER2-HER3 dimer expression in exosomes from HER2-positive patients receiving HER2 targeted therapies	Recruiting
NCT02862470	Observational	Anaplastic thyroid cancer, Follicular thyroid cancer	Prospective	Analysis of cancer-derived exosomes via lovastatin and vildagliptin treatments and prognostic study through urine exosomal markers	Active, not recruiting
NCT03102268	Observational	Cholangiocarcinoma	Prospective	Characterization of exosomal non-coding RNAs from cholangiocarcinoma patients before anticancer therapies	Unknown
NCT02393703	Observational	Pancreatic cancer	Prospective	Investigation of exosome mediated disease recurrence	Active, not recruiting
NCT03608631	Phase I	Metastatic pancreatic adenocarcinoma, Pancreatic ductal adenocarcinoma	Interventional	Study of the mesenchymal stromal cells-derived exosomes with KrasG12D siRNA (iExosomes) for pancreatic cancer patients having KrasG12D	Not yet recruiting
NCT03109873	Early phase I	Head and neck cancer	Randomized	Assessment of the effect of metformin hydrochloride on cytokines and exosomes in cancer patients	Completed
NCT01668849	Phase I	Head and neck cancer	Interventional	Investigation of the ability of plant-derived exosomes to prevent oral mucositis induced by combined chemotherapy and radiation	Active, not recruiting

## Data Availability

Not applicable.

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
