# Peer review of "Trends in Research on Exosomes in Cancer Progression and Anticancer Therapy"

_cancers, 2021, doi:10.3390/cancers13020326_

Round 1
Reviewer 1 Report
This article comprehensively reviews essential roles of exosomes in cancer biology research and the potential therapeutic applications with updated important references. Although it is well-written, there are several minor points to be revised.
Abstract
Line 23-24; Is the following sentence “Exosomes derived from various cells carry cargoes similar to their originator cells, making them unique compared to other extracellular vesicles.” true? Not only exosomes but also microvesicles have been considered to be drug carriers (e.g. Saari H, et al. 2020; Moore C, et al. 2017). The sentence should be revised not to mislead readers.
There is a huge number of abbreviations in this review article and some are not so popular for many researchers. Abbreviation list should be added in this article to help readers understand technical terms easily.
Line 69; EMT abbreviation is already described in line 65.
Line 96, 141, 142, 225, and Table 2 (Page 9); Please add what “NSF”, “ALIX”, “VPS4”, “PMN”, and “LAC” abbreviation are from.
Table 2 (Page 10); “gastric cancer” (Wang et al., 2017, Zhang et al., 2017) can be changed to GC.
7. Tumor-derived exosomes
Line 293-294; It should be mentioned this research finding is shown by using rat pancreatic adenocarcinoma line ASML.
Line 301; Regarding reference [84], please describe specific several miRNAs.
Line 327-330; In the section 7.5 (Exosomes and drug resistance), there is a lack of sentences/explanations after “Exosome mediated drug resistance has been evidenced with the following pathways:”. Besides, it is better to refer to neutralization of antibody-based drugs. To date, several studies have indicated that tumor-derived exosomes are involved into neutralizing antibody-based drugs and eventually decreased the overall effect of drugs (e.g. Ciravolo et al. 2012).
Line 329; The following sentence “Exosomes form a physical barrier against drug penetration and confers drug resistance by transfer of cargoes from resistant to sensitive cells” should be accompanied with not a review (Reference [31]) but original research papers as references.
8. Strategies against tumor-derived exosomes
Line 362-380; As “Suppression of exosome assembly release” and “uptake” are completely different, so it would be better to be discussed in the separate sections. In addition, though it is sure that there have been few reports regarding the disruption of exosome uptake by recipient cells, it is ideal to add more related articles.
9. Cancer management with exosomes
Line 407-416; In the section 9.1. (Exosomes as biomarkers), considering the clinical settings, it is hard to collect directly exosomes secreted from cancer cells. Therefore, please discuss which body fluid is suitable for biomarkers based on several research articles (e.g. ascites, blood, and saliva).
Line 455- 467; In the section 9.3. (Exosome-based RNA therapy), exosomal delivery with antisense miRNA oligonucleotides as well as miRNA/siRNA should be also mentioned (e.g. Kim G, et al. 2019).
Line 483-484; Please add references regarding the following sentence “The exosomal surface proteins regulate efficient drug delivery because of their involvement in exosomes uptake by the host cell.”
Line552-565; In the section 9.7. (Exosomes in clinical trials), it is helpful to include a new Table listing the clinical trials. In addition, please clarify that each trial mentioned in this section is Phase I, II, or III trial.
11. Conclusion
Line 616-623; I do not think these sentences are necessary in this review article.
References
Line 745; Please add doi:- of this reference.
Author Response
The authors of this manuscript express their sincere thanks to the reviewer for the critical assessment of this work. The authors have acted upon the recommendations of the reviewer which have resulted in a significant enhancement in the quality of this manuscript. All modifications incorporated in the manuscript are highlighted in red color font. A “point-by-point” response to each and every comment is outlined below.
Comment:
This article comprehensively reviews essential roles of exosomes in cancer biology research and the potential therapeutic applications with updated important references. Although it is well-written, there are several minor points to be revised.
Response:
The authors are highly thankful to the reviewer for the appreciation and suggestions. The concerns indicated by the reviewer were considered carefully and relevant information has been inserted where ever required.
Comment:
Abstract
Line 23-24; Is the following sentence “Exosomes derived from various cells carry cargoes similar to their originator cells, making them unique compared to other extracellular vesicles.” true? Not only exosomes but also microvesicles have been considered to be drug carriers (e.g. Saari H, et al. 2020; Moore C, et al. 2017). The sentence should be revised not to mislead readers.
Response:
Thank you for your suggestion. It is true that exosomes carry the features of their originator cells and their uniqueness lies in their mode of generation. This information has been inserted in abstract (page 1, lines 23 and 24).
Comment:
There is a huge number of abbreviations in this review article and some are not so popular for many researchers. Abbreviation list should be added in this article to help readers understand technical terms easily.
Response:
The authors are highly thankful for pointing out the need of an inclusive abbreviation list. So, we have provided a list of abbreviations used in the text after the conclusion section (pages 25-26).
Comment:
Line 69; EMT abbreviation is already described in line 65.
Response:
We greatly appreciate the reviewer’s close observation. The repetitive full form of EMT has been deleted (page 2, line 72).
Comment:
Line 96, 141, 142, 225, and Table 2 (Page 9); Please add what “NSF”, “ALIX”, “VPS4”, “PMN”, and “LAC” abbreviation are from.
Response:
We are extremely indebted to the learned reviewer for suggesting modifications. All necessary full forms for the abbreviations have been inserted in the revised text: ALIX (page 4, line 132), and VPS4 (page 4, line 151). NSF (page 3, lines 102 and 103) and PMN (page 7, line 233) abbreviated forms have been deleted as they were used only once in the manuscript. “LAC” has been replaced with “lung adenocarcinoma” (Table 2, page 10, reference 94).
Comment:
Table 2 (Page 10); “gastric cancer” (Wang et al., 2019, Zhang et al., 2017) can be changed to GC.
Response:
As per the suggestion, gastric cancer has been changed to “GC” at all relevant places throughout the manuscript. We have used the full form for the first time (page 9, line 313).
Comment:
Tumor-derived exosomes
Line 293-294; It should be mentioned this research finding is shown by using rat pancreatic adenocarcinoma line ASML.
Response:
This concern could not be addressed as we could not find any relevance for the line indicated by the reviewer.
Comment:
Line 301; Regarding reference [84], please describe specific several miRNAs.
Response:
We are grateful for this prudent suggestion and accordingly we have inserted the specific miRNAs, such as miR-25-3p, miR-130b-3p, and miR-425-5p (page 9, line 316-317). Since a lot of new information has been added, the reference no. has changed to [91].
Comment:
Line 327-330; In the section 7.5 (Exosomes and drug resistance), there is a lack of sentences/explanations after “Exosome mediated drug resistance has been evidenced with the following pathways:”. Besides, it is better to refer to neutralization of antibody-based drugs. To date, several studies have indicated that tumor-derived exosomes are involved into neutralizing antibody-based drugs and eventually decreased the overall effect of drugs (e.g. Ciravolo et al. 2012).
Response:
We are thankful to the erudite reviewer for the suggestion. For clarification the sentence “Exosome mediated drug resistance has been evidenced with the following pathways:” has been replaced by the sentence “Exosome-mediated drug resistance may be devised through trafficking of non-coding RNAs, drugs transporters and neutralization of antibody-based drugs which have been described in the following sections” (page 12, lines 345-347). The pathways involved in exosome-mediated drug resistance have been elaborately discussed under subheadings 7.5.1 (page 12, lines 349-362) and 7.5.2 (page 12, lines 364-376). Relevant information on neutralizing antibody-based drugs has also been provided (page 12, lines 373-376).
Comment:
Line 329; The following sentence “Exosomes form a physical barrier against drug penetration and confers drug resistance by transfer of cargoes from resistant to sensitive cells” should be accompanied with not a review (Reference [31]) but original research papers as references.
Response:
As per suggestion the reference 31 has been replaced with an original research paper which is now reference 104 (page 12, line 345).
Comment:
- Strategies against tumor-derived exosomes
Line 362-380; As “Suppression of exosome assembly release” and “uptake” are completely different, so it would be better to be discussed in the separate sections. In addition, though it is sure that there have been few reports regarding the disruption of exosome uptake by recipient cells, it is ideal to add more related articles.
Response
We are indebted to the reviewer for this extremely relevant suggestion. Accordingly, we have introduced two separate sections as follows:
8.1. Suppression of exosome biogenesis and trafficking in (page 12, lines 382-388; page 13, lines 391-393; page 13, lines 397-399; and page 13, lines 402-404)
8.2. Depletion of exosome uptake (page 13, lines 414-421).
Comment:
- Cancer management with exosomes
Line 407-416; In the section 9.1. (Exosomes as biomarkers), considering the clinical settings, it is hard to collect directly exosomes secreted from cancer cells. Therefore, please discuss which body fluid is suitable for biomarkers based on several research articles (e.g. ascites, blood, and saliva).
Response
Though liquid biopsy for identification of exosomal biomarker has tremendous potential in cancer diagnosis but the field is still in its infancy. There is no definite consensus for isolation of exosomes and therefore best suitable body fluid for exosome isolation is also under investigation. This information has been discussed under section 9.2. Exosomes as biomarkers (page14, lines 464-466 and page 14, lines 473-482).
Comment:
Line 455-467; In the section 9.3. (Exosome-based RNA therapy), exosomal delivery with antisense miRNA oligonucleotides as well as miRNA/siRNA should be also mentioned (e.g. Kim G, et al. 2019).
Response
We are thankful for this wonderful suggestion. Section 9.3 has been renumbered to 9.4. Relevant information regarding antisense miRNA oligonucleotides has now been added to this section (page16, lines 556-559, reference 175). siRNA based therapy has been already discussed (page 16, lines 549-556, references 171-174).
Comment:
Line 483-484; Please add references regarding the following sentence “The exosomal surface proteins regulate efficient drug delivery because of their involvement in exosomes uptake by the host cell.”
Response:
We have introduced a reference as suggested (page 17, lines 578-580, reference181).
Comment:
Line552-565; In the section 9.7. (Exosomes in clinical trials), it is helpful to include a new Table listing the clinical trials. In addition, please clarify that each trial mentioned in this section is Phase I, II, or III trial.
Response:
We are in absolute agreement with the reviewer. As suggested by the reviewer, a new table (Table 5) for exosomes in clinical trials has been incorporated at the end of section 9.8 (pages 22-23). The phase of each trial has been indicated as per the recommendation.
Comment:
- Conclusion
Line 616-623; I do not think these sentences are necessary in this review article.
Response:
We are extremely sorry for this inadvertent mistake and have removed the redundant sentences as indicated by the reviewer.
Comment:
References
Line 7 45; Please add doi:- of this reference.
Response:
Reference 45 has been replaced with Heusermann, W.; Hean, J.; Trojer, D.; Steib, E.; von Bueren, S.; Gra_-Meyer, A.; Genoud, C.; Martin, K.;Pizzato, N.; Voshol, J.; et al. Exosomes surf on philopodia to enter cells at endocytic hot spots, traffic with endosomes, and are targeted to the ER. J. Cell Biol. 2016, 213, 173–184, which is now reference 48 (page 5, line 194).
Additionally,
- The reference list has been modified as we have added several new references. Special attention is given to conform to the order of references and bibliographic style of the journal.
- The entire manuscript has been thoroughly checked and edited to ensure uniform style, organization, and quality.
On behalf of my co-authors, I once again express my sincere thanks to the erudite reviewer for the valuable suggestions and constructive input to improve the quality of our manuscript.
Reviewer 2 Report
In the review entitled “Emerging paradigm of exosomes in cancer progression and therapy” the authors described the different roles of exosomes in cancer research with a focus on the protumorigenic and antitumorigenic effect of the exosomes and therapeutics.
Unfortunately, trying to address so many aspects the review is on the not very interesting moreover, what was supposed to be the central focus was not fully addressed.
In my opinion, the first paragraphs up to and including the seventh should be revised and turned into an introduction on the multiple effects of exosomes in cancer. Paragraphs 8 to 10 are the most interesting and less well investigated but should be enriched by the most recent discoveries and applications with descriptions of the most relevant data.
In order to define exosomes, I strongly suggest referring to MISEV guideline 2014 and 2018 instead of the chosen references 2 and 3
Please check all the references, many of which are not correct or appropriate
(e.g.Ref 23 is not pertinent
Ref 24 is not pertinent, it is required a review or book chapter describing the role of exosomal proteins
Ref 25 did not refer to the priming of MVB budding
the review n.31 might be substitute with the original research manuscripts.
The same for the reviews 45-46-47-48 and 50.
Check reference line 197 probably the authors exchanged 48 for the 49
From line 240 please check references 57-58-59, do not like appropriate to the text.
End so on…)
Paragraph 7 with its subparagraph need to be reconsidered, they represent a surficial overview of the indicated topics, lacking to indicate the novelty. The large use of reviews as references supports this opinion.
The title of paragraph 7.2 have to be modified, the paragraph gives only a hint of possible epigenetic changes induced by exosomes
Overall I strongly suggest editing the review giving a more in-depth imprint to the aspect concerning the therapeutic approach
Author Response
The authors of this manuscript express their sincere thanks to the reviewer for the critical assessment of this work. The authors have acted upon the recommendations of the reviewer which have resulted in a significant enhancement in the quality of this manuscript. All modifications incorporated in the manuscript are highlighted in red color font. A “point-by-point” response to each and every comment is outlined below.
Comment:
In the review entitled “Emerging paradigm of exosomes in cancer progression and therapy” the authors described the different roles of exosomes in cancer research with a focus on the pro tumorigenic and antitumorigenic effect of the exosomes and therapeutics.
Unfortunately, trying to address so many aspects the review is on the not very interesting moreover, what was supposed to be the central focus was not fully addressed.
In my opinion, the first paragraphs up to and including the seventh should be revised and turned into an introduction on the multiple effects of exosomes in cancer. Paragraphs 8 to 10 are the most interesting and less well investigated but should be enriched by the most recent discoveries and applications with descriptions of the most relevant data.
Response:
We are thankful for reviewing our context. We agree with our learned reviewer that covering such a wide range of exosome associated topics had limited us with in depth discussion on a specific topic but on the other hand such a review would be helpful for scientists who have arrived new to the arena of exosome cancer biology. It would provide them with knowledge of both pro-tumorigenic and antitumorigenic effects of exosomes on a single platform. We have enriched the sections 8 to 9 with recent literature highlighting relevant information on exosomes (page 12, lines 382-388; page 13, lines 391-393; 397-399; 402-404; 414-421; page 14, lines 449-460, page 15, lines 487-490, 492-494, 499-502, and page 16, lines 517-526, 531-536). With due respect to our erudite reviewer, we would like to maintain our present review format where we have tried to cover all aspects of exosomes ranging from their implication in tumor progression to their anticancer therapeutic potential.
Comment:
In order to define exosomes, I strongly suggest referring to MISEV guideline 2014 and 2018 instead of the chosen references 2 and 3.
Response:
Thank you for the suggestion. Inclusion of MISEV guidelines 2014 and MISEV 2018 has been an excellent information which has been incorporated in our revised manuscript (page 1, line 43 to page 2, line 49, reference 2 and 3).
Comment:
Please check all the references, many of which are not correct or appropriate e.g. Ref 23 is not pertinent.
Response:
All references have been checked and corrected where ever necessary.
We have used a new reference [Li, Sp., Lin, Zx., Jiang, Xy. et al. Exosomal cargo-loading and synthetic exosome-mimics as potential therapeutic tools. Acta Pharmacol Sin 39, 542–551 (2018). https://doi.org/10.1038/aps.2017.178] to replace the old reference 23 which is now reference 25 (page 3, line 104).
Comment:
Ref 24 is not pertinent, it is required a review or book chapter describing the role of exosomal proteins.
Response:
We are intrigued by the meticulous reference check of the reviewer and have changed the previous reference 24 with a new one [Zhang, Y., Liu, Y., Liu, H. et al. Exosomes: biogenesis, biologic function and clinical potential. Cell Biosci 9, 19 (2019). https://doi.org/10.1186/s13578-019-0282-2] which is now reference 26 (page 3, line 109).
Comment:
Ref 25 did not refer to the priming of MVB budding
Response: Reference 25 has been replaced with a new reference [Souza-Schorey, C.; Schorey, J.S. Regulation and mechanisms of extracellular vesicle biogenesis and secretion. Essays Biochem. 2018, 62, 125–133, doi:10.1042/EBC20170078] which is now reference 27 (page 4, line 130). Another reference [Antonyak, M.A.; Cerione, R.A. Microvesicles as mediators of intercellular communication in cancer. Methods Mol. Biol. 2014, 1165, 147–173, doi:10.1007/978-1-4939-0856-1_11] as reference 28 (page 4, line 130).
Comment:
the review n.31 might be substitute with the original research manuscripts.
Response:
The reference no. 31 previously appeared in four places out of which its presence in three positions (in section 5 and section 7.3) remained unchanged as they referred to a bouquet of information as mentioned in this important publication. In section 7.5 (page 12, line 345), the reference no. 31 has been replaced with a new reference no.104 which is an original research paper [Qin, X.; Yu, S.; Zhou, L.; Shi, M.; Hu, Y.; Xu, X.; Shen, B.; Liu, S.; Yan, D.; Feng, J. Cisplatin-resistant lung cancer cell–derived exosomes increase cisplatin resistance of recipient cells in exosomal miR-100–5p-dependent manner. Int. J. Nanomedicine 2017, 12, 3721–3733, doi:10.2147/IJN.S131516].
Comment:
The same for the reviews 45-46-47-48 and 50.
Response:
Reference 45 has been replaced with Heusermann, W.; Hean, J.; Trojer, D.; Steib, E.; von Bueren, S.; Gra_-Meyer, A.; Genoud, C.; Martin, K.;Pizzato, N.; Voshol, J.; et al. Exosomes surf on philopodia to enter cells at endocytic hot spots, traffic with endosomes, and are targeted to the ER. J. Cell Biol. 2016, 213, 173–184, which is now reference 48 (page 5, line 194).
Reference 46 has been replaced with Tian, T.; Zhu, Y.L.; Zhou, Y.Y.; Liang, G.F.; Wang, Y.Y.; Hu, F.H.; Xiao, Z.D. Exosome uptake through clathrin-mediated endocytosis and macropinocytosis and mediating miR-21 delivery. J. Biol. Chem. 2014, 289, 22258–22267, doi:10.1074/jbc.M114.588046, which is now reference 49 (page 5, line 201).
Reference 47 has been replaced with Nabi IR, Le PU. Caveolae/raft-dependent endocytosis. J Cell Biol. 2003;161(4):673-677. doi:10.1083/jcb.200302028, which is now reference 50 (page 5, line 203).
Reference 48 has been replaced with Gonda A, Kabagwira J, Senthil GN, Wall NR. Internalization of Exosomes through Receptor-Mediated Endocytosis. Mol Cancer Res. 2019 ;17(2):337-347. doi: 10.1158/1541-7786, which is now reference 51 in (page 5, line 206).
Reference 49 has been replaced with Joshi BS, de Beer MA, Giepmans BNG, Zuhorn IS. Endocytosis of Extracellular Vesicles and Release of Their Cargo from Endosomes. ACS Nano. 2020;14(4):4444-4455. doi: 10.1021/acsnano.9b10033, which is now reference 52 (page 5, line 207).
Reference 50 has been replaced with Andreu Z, Yáñez-Mó M. Tetraspanins in extracellular vesicle formation and function. Front Immunol. 2014; 5:442. Published 2014 Sep 16. doi:10.3389/fimmu.2014.00442, which is now reference 53 (page 5, line 207).
Comment:
Check reference line 197 probably the authors exchanged 48 for the 49.
Response:
Thank you for this comment though there was no exchange of references. However, reference 48 has been replaced with reference 51 [Gonda A, Kabagwira J, Senthil GN, Wall NR. Internalization of Exosomes through Receptor-Mediated Endocytosis. Mol Cancer Res. 2019 ;17(2):337-347. doi: 10.1158/1541-7786] (page 5 line 206). Similarly, reference 49 has been replaced with reference 52 [Joshi BS, de Beer MA, Giepmans BNG, Zuhorn IS. Endocytosis of Extracellular Vesicles and Release of Their Cargo from Endosomes. ACS Nano. 2020;14(4):4444-4455. doi: 10.1021/acsnano.9b10033] (page 5, line 207).
Comment:
From line 240 please check references 57-58-59, do not like appropriate to the text.
End so on…)
Response:
The references have been cross-checked and have been found to be correct.
Comment:
Paragraph 7 with its subparagraph need to be reconsidered, they represent a surficial overview of the indicated topics, lacking to indicate the novelty. The large use of reviews as references supports this opinion.
Response:
This is an excellent suggestion and accordingly section 7 have been revised (page 7, line 276 to page 8, line 278; page 8, lines 283-284; page 8, lines 288-290; page 9, lines 314-315; and page 9, lines 323-325). Keeping in mind the objectives of the review, which aimed to cover wide range of tumorigenic responses exerted by exosomes as well as the antitumor implications of exosomes, we have designed concise subsections under a main section. However, in certain circumstances, citations of reviews have been executed to incorporate multiple information in a compact mode.
Comment:
The title of paragraph 7.2 have to be modified, the paragraph gives only a hint of possible epigenetic changes induced by exosomes.
Response:
We believe the reviewer has made a terrific point. The title of subsection 7.2 has been modified to “Exosomal miRNA-mediated cancer promotion” (page 7, line 254).
Comment:
Overall I strongly suggest editing the review giving a more in-depth imprint to the aspect concerning the therapeutic approach.
Response:
We are thankful to the reviewer for the suggestion and have done rigorous modifications, emphasizing therapeutic role of exosomes in cancer management.
Additionally,
- The reference list has been modified as we have added several new references. Special attention is given to conform to the order of references and bibliographic style of the journal.
- The entire manuscript has been thoroughly checked and edited to ensure uniform style, organization, and quality.
On behalf of my co-authors, I once again express my sincere thanks to the erudite reviewer for the valuable suggestions and constructive input to improve the quality of our manuscript
Reviewer 3 Report
Emerging paradigm of exosomes in cancer progression and therapy.
By Dona Sinha et al.
This is a comprehensive review of exosomes in cancer progression and therapy and yes the authors touched all areas that are under intense investigation. Having said that, the title suggests an emerging paradigm in the field as in paradigm shift. To an investigator who works in the field of extracellular vesicles the title suggests a new paradigm that warrants serious attention. However, the review is typical of most Extracellular Vesicle Reviews that have appeared in the literature. Most reviews talk about miR's in exosomes that regulate various cancer progression processes such as migration and invasion without pointing to the the specific proteins that are affected and hence highly descriptive. Studies suggest various uptake pathways for exosomes such as phagocytosis but short on details as to which is relevant in vivo. The review should be extensively re-organized to emphasize the emerging paradigm in exosome research and that is the mechanistic views of the roles of exosomes in cancer. For example it is now becoming clear that miR's are not even in exosomes per se but in non-membrane nanoparticles where they interact with Argos.
Apart from the fact that the review is not fundamentally different from other comprehensive reviews of exosomes, there are minor issues that the authors need to address.
1) Check the last word on line 92 for accuracy.
2) Line 198, it should be exosomes and not exsosomes.
3) Line 199, This process is followed...and not This process followed.
4) line 233, metastatic properties and not metastatic property.
5) Line 234, culminate in greater release and not culminate greater release.
6) Line 246, change to "Epigenetic regulation of cancer by exosomes"
Author Response
The authors of this manuscript express their sincere thanks to the reviewer for the critical assessment of this work. The authors have acted upon the recommendations of the reviewer which have resulted in a significant enhancement in the quality of this manuscript. All modifications incorporated in the manuscript are highlighted in the red color font. A “point-by-point” response to each and every comment is outlined below.
Comment:
This is a comprehensive review of exosomes in cancer progression and therapy and yes the authors touched all areas that are under intense investigation. Having said that, the title suggests an emerging paradigm in the field as in paradigm shift. To an investigator who works in the field of extracellular vesicles the title suggests a new paradigm that warrants serious attention. However, the review is typical of most Extracellular Vesicle Reviews that have appeared in the literature. Most reviews talk about miR's in exosomes that regulate various cancer progression processes such as migration and invasion without pointing to the the specific proteins that are affected and hence highly descriptive. Studies suggest various uptake pathways for exosomes such as phagocytosis but short on details as to which is relevant in vivo. The review should be extensively re-organized to emphasize the emerging paradigm in exosome research and that is the mechanistic views of the roles of exosomes in cancer. For example, it is now becoming clear that miR's are not even in exosomes per se but in non-membrane nanoparticles where they interact with Argos.
Apart from the fact that the review is not fundamentally different from other comprehensive reviews of exosomes, there are minor issues that the authors need to address.
Response:
We are extremely thankful to the reviewer for the thought-provoking comments and excellent recommendations. We have tried to work on all suggestions made by three reviewers and have incorporated major changes throughout manuscript. Regarding mechanistic insights into the protumorigenic effects of exosomes, we would like to mention that the entire table 2 has been specifically constructed to indicate the specific molecules by which the exosomal cargoes mediate the pro-tumorigenic changes in the target cells. Similarly, table 3 and table 4 have depicted exosome-mediated anticancer therapeutic potential and the associated mechanisms. In conjunction with succinct summary of relevant information presented in the tables, various sections of our manuscript are also loaded with mechanistic underpinnings of exosomal effects, such as Section 7.5 (page 12, lines 355-382). In order to comply with our reviewer’s suggestion, we have modified the title of the review to “Trends in research on exosomes in cancer progression and anticancer therapy. Throughout the review, we have tried to cover major aspects of protumorigenic and antitumorigenic potential of exosomes in order to present a critical and comprehensive knowledge for researchers who have just stepped in the world of exosome research. The novelty of the review lies in the fact that instead of perceiving specific protumorigenic effects of exosomes or therapeutic potential of exosomes, the present review has tried to decipher the entire repertoire of exosomes, which promote tumor growth and can be targeted for antitumor therapy.
Comment
Check the last word on line 92 for accuracy.
Response:
The correct spelling of “retrotransposon” has been used (page 3, line 99).
Comment:
Line 198, it should be exosomes and not exsosomes.
Response:
The spelling of the word “exosomes” have been corrected (page 5, line 207).
Comment:
Line 199, This process is followed...and not This process followed.
Response:
The sentence has been modified as suggested (page 6, line 208).
Comment:
line 233, metastatic properties and not metastatic property.
Response:
The suggested change has been inserted (page 7, line 241).
Comment:
Line 234, culminate in greater release and not culminate greater release.
Response:
The sentence has been modified as suggested (page 7, line 242).
Comment:
Line 246, change to "Epigenetic regulation of cancer by exosomes"
Response:
We sincerely appreciate the suggestion. Reviewer 2 also suggested a different title as the paragraph gives only a hint of possible epigenetic changes induced by exosomes. In order to strike a balance between suggestions from both the reviewers, we have used this tile “Exosomal miRNA-mediated cancer promotion” (page 7, line 254).
Additionally,
- The reference list has been modified as we have added several new references. Special attention is given to conform to the order of references and bibliographic style of the journal.
- The entire manuscript has been thoroughly checked and edited to ensure uniform style, organization, and quality.
On behalf of my co-authors, I once again express my sincere thanks to the erudite reviewer for the valuable suggestions and constructive input to improve the quality of our manuscript
Round 2
Reviewer 2 Report
The revised manuscript can be considered for publication.
Reviewer 3 Report
This revised version is now much improved and the requested revisions thoroughly addressed.